# CPMF-Net: Multi-Feature Network Based on Collaborative Patches for Retinal Vessel Segmentation

**DOI:** 10.3390/s22239210

**Published:** 2022-11-26

**Authors:** Wentao Tang, Hongmin Deng, Shuangcai Yin

**Affiliations:** School of Electronics and Information Engineering, Sichuan University, Chengdu 610065, China

**Keywords:** fundus image, vessel segmentation, training methods, self-attention, channel attention

## Abstract

As an important basis of clinical diagnosis, the morphology of retinal vessels is very useful for the early diagnosis of some eye diseases. In recent years, with the rapid development of deep learning technology, automatic segmentation methods based on it have made considerable progresses in the field of retinal blood vessel segmentation. However, due to the complexity of vessel structure and the poor quality of some images, retinal vessel segmentation, especially the segmentation of Capillaries, is still a challenging task. In this work, we propose a new retinal blood vessel segmentation method, called multi-feature segmentation, based on collaborative patches. First, we design a new collaborative patch training method which effectively compensates for the pixel information loss in the patch extraction through information transmission between collaborative patches. Additionally, the collaborative patch training strategy can simultaneously have the characteristics of low occupancy, easy structure and high accuracy. Then, we design a multi-feature network to gather a variety of information features. The hierarchical network structure, together with the integration of the adaptive coordinate attention module and the gated self-attention module, enables these rich information features to be used for segmentation. Finally, we evaluate the proposed method on two public datasets, namely DRIVE and STARE, and compare the results of our method with those of other nine advanced methods. The results show that our method outperforms other existing methods.

## 1. Introduction

As important sensory organs, more than 70% [1] of information to the brain is provided via eyes, which are most key channels for people to perceive the world. However, affected by some retinal diseases, such as diabetic retinopathy, glaucoma, cataract and so on, many people are at a moderate or above visual risk. For these patients, if they cannot get timely medical intervention, they might suffer from deterioration of vision and even blindness with the development of these diseases. Therefore, timely diagnosis and treatment are very significant for relevant patients. According to medical researches [2,3], the appearance of retinal diseases is often accompanied by the morphological changes in retinal vessels. Clinical doctors usually find diseases by extensive experience through observing these changes. Therefore, how to automatically segment retinal vessels from fundus images by computer is of significance in assisting clinical doctors to diagnosis these diseases. In recent years, with the gradual demonstration of its potential application, deep learning has been widely used in the field of computer vision, including the segmentation of retinal vessels. Under continuous explorations, considerable results have been achieved by deep-learning-based segmentation methods. However, due to the complex and diverse structures of retinal vessels, the low contrast between vessels and background in fundus images, it is often difficult to correctly segment the densely connected fine vessels in retinal vessels, which remains a challenge. This paper is focused on two difficult problems in the field of retinal vessel segmentation, one is how to better segment the fine blood vessels in the fundus image, and the other is how to choose the network training mode, end-to-end mode or patch-based mode, so as to better reduce the occupation of computing resources and the loss of image information. In order to address the above two problems, this paper proposes a multi-feature collaborative segmentation network and collaborative patch training strategy for retinal vessel segmentation. On the one hand, the fusion structure of the pre-segmentation network and main-segmentation network can effectively improve the segmentation details of vessel structure, especially the fine vessel structure. On the other hand, the designed collaborative patch training strategy not only has the advantages of simple vessel structure and less computational resources of the patch-based training method but also effectively reduces the information loss caused by patch extraction. The contributions of this paper can be summarized as follows:1.A multi-feature segmentation network is proposed for retinal vessel segmentation. The two-level sub-networks complete the pre-segmentation and main-segmentation tasks, respectively. The pre-extraction of basic information of blood vessels in pre-segmentation and the cooperation of multiple information features in main-segmentation provide a lot of effective information for blood vessel segmentation, which improves the segmentation accuracy of the network, especially in the face of difficult blood vessels.2.A collaborative patch training strategy is designed to reduce the information loss in the patch-based method. On the basis of patches, the segmentation method combining one small patch with simple vessel structure and five large patches with global information not only effectively retains the advantages of patch-based method but also effectively reduces the information loss caused by patch extraction.3.An adaptive coordinate attention module is designed to extract the direction information of blood vessels. This module provides the model with very helpful vessel orientation information for vessel structure segmentation and improves the vessel continuity in the model segmentation results.4.A gated self-attention module suitable for retinal image segmentation task is designed. The self-attention module integrated into the main-segmentation network can alleviate the local dependence of convolution operation and help the network to obtain long-distance dependence.

## 2. Related Work

### 2.1. Network Structure for Retinal Vessel Segmentation

In the past decade, accompanied by the rapid development of deep learning, convolutional neural networks (CNNs) have also been widely carried out in the field of image segmentation, including retinal vessel segmentation. Compared with retinal vessel segmentation methods based on classical classifiers such as support vector machine (SVM) and k-nearest neighbor (KNN), the retinal vessel segmentation methods [4,5,6] based on convolutional neural network can more effectively extract image features from fundus images. Motivated by the great potential of convolutional neural networks on image segmentation, researchers did a lot of work in this area and proposed quite valuable structures, such as fully convolutional networks (FCNs) [7]. By extending the FCNs, a network with U-shaped encoder–decoder architecture (U-Net) was proposed by Ronne-berger et al. [8]. The “skip-connection” structure between the downsampling encoder and the upsampling decoder in the U-Net could combine the semantic information from the high-level features with the spatial information from the low-level features to jointly complete the segmentation. The great potential of U-Net in image segmentation made it a basic backbone widely used in the field of medical image segmentation, including retinal vessel segmentation. Fu et al. [9] applied the framework of multi-scale convolutional neural network to the segmentation of retinal vessels. Through the rich hierarchical representation and the application of conditional random fields, the segmentation accuracy of retinal vessels was improved. Mo et al. [10] developed a deep supervised FCN, which improved the clinical applicability of the network by using the multi-level characteristics of the deep network. Guo et al. [11] designed a channel attention residual U-Net, which improved the extraction ability of retinal vessel features by adding modified efficient channel attention module to the U-Net. The above methods have improved the segmentation results of retinal vessels to a certain extent, but there are still deficiencies in the complex but very important task of small vessel segmentation. A multi-feature segmentation network is proposed in this paper. Through a cascade of stage segmentations and the integration of multiple effective attention modules, the network was effectively improved in the segmentation of fine blood vessels.

### 2.2. Training Method of the Model

In general, the training methods of retinal vessel segmentation network based on a convolutional neural network can be roughly divided into end-to-end and patch-based methods, where the end-to-end method has attracted the attention of many researchers because of its simple and stable characteristics in training. However, due to the high resolution of fundus images and limited computing resources, the end-to-end method that requires the network to learn the whole image usually includes downsampling of the image, but this would undoubtedly lose the spatial information that is very important for segmentation. Therefore, patch-based training has become a more widely used method in recent years because of its advantages such as less computational resources and simpler vessel morphology. Liskowski et al. [12] used deep learning to train the network on the large dataset via data enhancement and complete the segmentation of blood vessels. Wu et al. [13] proposed a multi-scale network following network (MS-NFN) model to address the problem of accurate segmentation of blood vessels, in which different upper poolings and lower poolings were used to complete the segmentation of blood vessels. After that, they further proposed an NFN+ model [14] based on network following network, which obtained rich samples for network training through patch extraction and data enhancement of images. Nevertheless, most of the above patch-based methods did not pay enough attention to the information loss caused by patch extraction. In order to eliminate the negative impacts of information loss on blood vessel segmentation, in this paper, a collaborative patch training strategy is designed. By extracting information from the large patches based on the target region patch during segmentation, the information loss of the target patch is effectively alleviated.

## 3. Methods

The flow chart of the multi-feature segmentation method based on collaborative patches proposed in this paper is shown in Figure 1, where the collaborative patch training strategy is used to effectively reduce the information loss of patches through the transmission of associated information between collaborative patches and improve the segmentation accuracy, while maintaining low computational resources. Moreover, the application of a multi-feature segmentation network further improves the accuracy of the segmentation for retinal vessels, especially for capillaries, through hierarchical structure and aggregation of multiple information features.

### 3.1. Multi-Feature Segmentation Network

The architecture of the multi-feature segmentation network designed in this paper is shown in Figure 2. The network is composed of a pre-segmentation sub-network, main-segmentation sub-network and edge extraction branch. Among them, the pre-segmentation sub-network is responsible for the rough segmentation of the image, the edge extraction branch is responsible for the extraction of blood vessel edges, and the main-segmentation sub-network is responsible for the fine segmentation of the image. Both segmentation sub-networks are based on the framework of U-Net, but for the more fine segmentation of vessel structure, we integrate an adaptive coordinate attention module for perceiving the direction information and a gated self-attention module for reducing the local dependence of convolution in the second cascade network. In addition, considering the significance of vessel edge information in vessel segmentation, we also design a learnable edge extraction branch to extract the edge information of vessel structure from fundus images, which is next sent to the main-segmentation sub-network to improve the segmentation results.

### 3.2. Adaptive Coordinate Attention Module

As one of the ways to enhance the model, an attention module has been widely used in image segmentation tasks. The attention module integrated into the network can improve the extraction of effective information, as well as the interpretability of network prediction by observing the attention weight value. Inspired by research [15], we design an adaptive coordinate attention module, as shown in Figure 3. The design of this module is the application of classical channel attention in the retinal segmentation task. The main motivation of the module design is listed in the following three aspects. First, the direction information of blood vessels is very useful for segmentation because the retinal vessel structure has choroidal characteristics. In order to better obtain the direction information, we adopt one-dimensional pooling along the horizontal and vertical directions separately to encode the information, which makes the module very sensitive to the direction of blood vessels and improves the module extraction of the direction information of blood vessels. Second, in the task of retinal vessel segmentation, the detection and segmentation of micro-vessels remains a very challenging task. However, many existing attention modules only use the average pooling method to encode information, which blurs the difference between the capillaries and the background, resulting in the capillaries being easily regarded as the background area. Therefore, we add the maximum pooling code on the basis of average pooling, making the boundaries between the capillary region and the background region more distinct and improving the module’s perception of the capillary region. Third, at present, in order to reduce the information loss caused by pooling operation, many channel attention modules introduce a variety of weights into different poolings. Even so, the simple addition of weights without considering the sample situation may aggravate the loss of information. Therefore, we use a learnable adaptive weight when combining the two pooling coding methods, so that the module can choose the best combination method through learning. Specifically, for an input $T\in R^{C\times H\times W}$, we use one-dimensional pooling to encode each channel information along the horizontal and vertical dimensions. Therefore, the output of the *c*-th channel at the width *w* can be defined as:(1)$$k_{c}^{w_{a}}\left(w\right)=\frac{1}{H}\sum_{0\leq i\leq H-1}T_{c}\left(i,w\right)$$
(2)$$k_{c}^{w_{m}}\left(w\right)=max\left(\left[T_{c}\left(0,w\right),\ldots ,T_{c}\left(H-1,w\right)\right]\right)$$

Similarly, the output of the *c*-th channel at the height *h* can be defined as:(3)$$k_{c}^{h_{a}}\left(h\right)=\frac{1}{W}\sum_{0\leq j\leq W-1}T_{c}\left(h,j\right)$$
(4)$$k_{c}^{h_{m}}\left(h\right)=max\left(\left[T_{c}\left(h,0\right),\ldots ,T_{c}\left(h,W-1\right)\right]\right)$$

For the generated feature, we splice it into the two directional features in the two branches, respectively, and then feed them to the convolution transform function $F_{1}$ shared by the two branches to obtain the intermediate features $z_{a}$ and $z_{m}$. They can be expressed as:(5)$$z_{a}=\delta (F_{1}([k_{a}^{h},k_{a}^{w}]))$$
(6)$$z_{m}=\delta (F_{1}([k_{m}^{h},k_{m}^{w}]))$$
where [,] represents the concatenation operation on the spatial dimension, and δ represents a non-linear activation function; the two intermediate features $z_{a}\in R^{\left[C/r\right]\times \left(H+W\right)}$ and $z_{m}\in R^{\left[C/r\right]\times \left(H+W\right)}$ are the features of horizontal and vertical spatial information encoded by different coding methods in the two branches, respectively. Where *r* is the reduction ratio, and [] represents rounding operation. Then, the tensors in the two branches are split into two independent tensors based on the corresponding dimension information. The resulting tensor is the merge of two tensors in the same direction from two branches by adaptively adjusting weights. The resulting horizontal and vertical tensors are transformed to the same number of channels as the input through 1 × 1 transformation functions $F_{h}$ and $F_{w}$, respectively. The output can be formulated as:(7)$$z^{w}=g_{1}*z_{a}^{w}+g_{2}*z_{m}^{w}$$
(8)$$z^{h}=g_{1}*z_{a}^{h}+g_{2}*z_{m}^{h}$$
(9)$$f_{h}=\delta (F_{h}(z^{h}))$$
(10)$$f_{w}=\delta \left(F_{w}\left(z^{w}\right)\right)$$
where $g_{1}$ and $g_{2}$ are adaptive weights that can be learned in the two branches, respectively, and δ is the sigmoid function. Finally, input $T_{c}$ obtains the output $y_{c}$ of the module under the action of two directional weights $f_{h}$ and $f_{w}$, and the output $y_{c}$ can be formulated as:(11)$$y_{c}\left(i,j\right)=T_{c}\left(i,j\right)\times f_{c}^{h}\left(i\right)\times f_{c}^{w}\left(j\right)$$

### 3.3. Gated Self-Attention Module

In recent years, Transformer has been brilliant in many fields including retinal vessel segmentation. However, if we want to apply Transformer to the field of retinal vessel segmentation, we have to face two problems. One is the high resolution of fundus images, which means a huge amount of computation in the calculation of self-attention. The second is that the training of Transformer is more prone to be unsatisfactory, especially the training of relative position bias. Inspired by research [16], we design a gated self-attention module from the above two problems. By modifying the traditional self-attention in the above two aspects, the gated self-attention module is more suitable for retinal vessel segmentation tasks. On the one hand, we reduce the negative impact of non-ideal relative position bias on the segmentation results by adaptive control of the relative position bias. On the other hand, our self-attention calculation is carried out in the feature map after cutting, which greatly reduces the amount of calculation brought by calculating the autocorrelation coefficient. At the same time, through the cooperation of different segmentation methods, we reduce the loss of inter-patch dependence brought by segmentation. The structure of the gated self-attention module is shown in Figure 4. The module is composed of a pacth self-attention block, LayerNorm (LN) layer, residual connection and 2-layer MLP. The output of the module can be formulated as:(12)$${\tilde{t}}^{n}=PSA(LN(t^{n-1}))+t^{n-1}$$
(13)$$t^{n}=MLP(LN({\tilde{t}}^{n}))+{\tilde{t}}^{n}$$
(14)$${\tilde{t}}^{n+1}=PSA\left(LN\left(Shift\left(t^{n}\right)\right)\right)+Shift\left(t^{n}\right)$$
(15)$$t^{n+1}=MLP\left(LN\left({\tilde{t}}^{n+1}\right)\right)+{\tilde{t}}^{n+1}$$
where $t^{n-1}$ is the input of the module, $t^{n+1}$ is the output of the module and PSA is the patch self-attention we designed. Similar to other self-attention mechanisms, the detail of PSA is shown in Figure 4. The patch self-attention output is:(16)$$Attention\left(Q,K,V\right)=Softmax\left(\frac{QK^{t}}{\sqrt{d}}+g_{a}*B\right)$$
where Q,K and $V\in R^{M^{2}\times d}$ are query, key and value matrices, respectively, and $M^{2}$ and *d* represent the number of blocks and the dimension of query or key matrix, respectively. $B\in R^{M^{2}\times M^{2}}$ represents the relative position bias matrix, and its value comes from the deviation matrix $\tilde{B}\in R^{\left(2M+1\right)\times \left(2M-1\right)}$ [17,18].

### 3.4. Collaborative Patch Training Strategy

Patch-based methods usually require less computing resources than end-to-end methods, because the training model uses the extracted patches rather than the whole image. However, the process of extracting patches from the whole image will inevitably lead to the loss of information due to the separation of pixels at the edge of the patch. The loss of this part of information is not beneficial for the segmentation of vessel structures. Based on the above analysis, in order to reduce the negative impact of information loss on vessel segmentation, we design a collaborative patch training strategy. In addition to one small patch that completely coincides with the target region, in this strategy, the features of five additional large patches composed of target regions and part of its neighborhoods are also extracted. Compared with the small patch, the vascular structure in each of the large patches is more complete but also more complex, so their segmentation features are more complete, although the segmentation details are poor. A more complete vascular structure means that their features contain the correlation between the blood vessels in the target region and the blood vessels in the neighborhood. Therefore, we can obtain the associated information between the target region and the neighborhood by extracting the structural relationship between the two regions. The obtained associated information will participate in the segmentation of the small patch and promote the segmentation of the small patch by making up the lost associated information of the small patch. Through patch collaborative extraction and associated information transmission, the lost information can be compensated for to help the segmentation of vessel structure in this paper.

#### 3.4.1. Patch Collaborative Extraction

Different from other patch-based methods, as shown in Figure 1, we not only extract a small patch that completely coincides with the target area but also extract five additional large patches that contain the target area and its surrounding areas. Among them, a small patch with simple and easily segmented vessel structure provides the vessel structure information, and five large patches with more complete information supplement the associated information lost due to pixel separation on the edge of the target area. The setting of five collaborative patches can ensure that there are enough associated information sources in the target area no matter where the target area is located in the image. Specifically, the small patch *S* completely coincides with the target area and is used to extract the main vessel structure information for the network. Large patches $L_{1}$, $L_{2}$, …, $L_{5}$ are five combinations of the target area and surrounding areas, respectively, which are used to supplement the associated information lost due to pixel separation.

#### 3.4.2. Associated Information Transmission

In order to effectively extract and transmit the associated information in the large patches $L_{1}$, $L_{2}$, …, $L_{5}$, we design an associated information fusion module and an associated information correction network to transmit the large patch information in two stages to help segment the vessel structure of the target area. The structure of the associated information fusion module and the associated information correction network is shown in the Figure 5. The associated information fusion module is integrated between the pre-segmentation sub-network and the main-segmentation sub-network. It will extract the lost associated information due to pixel separation from the pre-segmentation results of the five large patches and adaptively combine the associated information in the five large patches through the internal attention module. Then, the resulting associated information features combine with the pre-segmentation results of the small patch into the main-segmentation stage.

Specifically, for each of the pre-segmentation features $f_{p}^{l_{1}}$, $f_{p}^{l_{2}}$, …, $f_{p}^{l_{5}}$ in five large patches, the first is a 3 × 3 convolution to obtain the associated information. Then, the associated information is combined adaptively through the attention module after splicing. Finally, the associated information is transformed by a 1 × 1 convolution to extract the associated information features $f_{p}^{l_{t}}$ of the target region. $f_{p}^{l_{t}}$ can be formulated as:(17)$$f_{p}^{l_{t}}=F_{1\times 1}\left(CA\left(Concat\left(F_{3\times 3}\left(f_{p}^{l_{1}},f_{p}^{l_{2}},\ldots ,f_{p}^{l_{5}}\right)\right)\right)\right)$$
where, CA represents the channel attention operation, $F_{1\times 1}$ and $F_{3\times 3}$ represent the convolution transformations of 1 × 1 and 3 × 3, respectively, and Concat represents the concatenation operation. Similarly, for the features $f_{m}^{l_{1}}$, $f_{m}^{l_{2}}$, …, $f_{m}^{l_{5}}$ and $f_{m}^{s}$ generated by the large patches and a small patch in the main-segmentation stage, the associated information in $f_{m}^{l_{1}}$, $f_{m}^{l_{2}}$, …, $f_{m}^{l_{5}}$ will be extracted again through the associated information correction network and used to correct and supplement the information of the small patch. Specifically, for each of $f_{m}^{l_{1}}$, $f_{m}^{l_{2}}$, …, $f_{m}^{l_{5}}$, firstly, a 3 × 3 convolution transform is used to extract each of the information features ${\tilde{f_{m}}}^{l_{1}}$, ${\tilde{f_{m}}}^{l_{2}}$, …, ${\tilde{f_{m}}}^{l_{5}}$, respectively. On the one hand, the extracted information features are used to generate large segmentation results through a 1 × 1 convolution transform. On the other hand, ${\tilde{f_{m}}}^{l_{1}}$, ${\tilde{f_{m}}}^{l_{2}}$, …, ${\tilde{f_{m}}}^{l_{5}}$ will be cut into the same size as $f_{m}^{s}$ according to the relationship with the target area, and after splicing, the channel attention module will adaptively combine them to generate correction information feature $f_{m}^{l_{t}}$ required by the small patch. Finally, the small patch feature $f_{m}^{s}$ and the correction information feature $f_{m}^{l_{t}}$ are sent into two convolution layers to generate the vessel segmentation results of the small patch. The correction information feature $f_{m}^{l_{t}}$ and the segmentation result $S_{s}$ of the small patch can be formulated as:(18)$$f_{m}^{l_{t}}=CA\left(cut\left(F_{1}\left(f_{m}^{l_{1}},f_{m}^{l_{2}},\ldots ,f_{m}^{l_{5}}\right)\right)\right)$$
(19)$$S_{s}=F_{2}\left(F_{1}\left(f_{m}^{s},f_{m}^{l_{t}}\right)\right)$$
where, CA represents the channel attention operation, $F_{1}$ and $F_{2}$ represent 3 × 3 and 1 × 1 convolution transforms, respectively, and cut represents the cutting operation.

## 4. Experiments

### 4.1. Dataset

Two public datasets are used to evaluate the proposed multi-feature collaborative segmentation network and collaborative patch training strategy. The information of the datasets is shown in Table 1.

The DRIVE [19] dataset consists of 40 RGB fundus images, in which 20 images are selected as training sets and 20 images are selected as test sets. The resolution of each image is 565 × 584. For the test set, we use the first manual annotation to evaluate the performance of segmentation.

The STARE [20] data set consists of 20 RGB fundus images, including 10 pathological images and 10 normal images. The resolution of each image is 700 × 605. Among them, 16 images are selected as the training set and 4 images are selected as the test set.

For DRIVE datasets, each image has a corresponding field of view (FOV) mask. However, there is no corresponding FOV mask for the image in the STARE dataset, so we use the method proposed by Marin et al. [21] to generate the mask for the image. All index calculations of our experiments in this paper only consider the pixels in the FOV mask.

### 4.2. Pre-Processing

Considering that the low quality of some original fundus images may hinder the segmentation of vessels, some pre-processing methods are adopted to improve the quality of the fundus images. As shown in Figure 6, after graying, contrast-limited adaptive histogram equalization (CLAHE) and gamma correction, the display quality of blood vessel structure in the fundus image has been significantly improved. In order to avoid over-fitting and improve the generalization ability of the model, a series of data enhancement methods are also adopted to increase the amount of data before training. The enhancement methods include random rotation operation, random erase operation, up and down flip operation, left and right flip operation, etc.

### 4.3. Evaluation Metrics

In this section, a series of indicators are used to quantitatively and comprehensively evaluate the proposed model, including accuracy (ACC) reflecting the correct classification of pixels, sensitivity (SE) reflecting the correct detection of blood vessels, specificity (SP) reflecting the correct detection of background, F1-score (F1) reflecting the recall and accuracy and area under receiver operating characteristic curve (AUC), reflecting the comprehensive performance of multiple aspects. These indicators can be formulated as:(20)$$ACC=\frac{TP+TN}{TP+TN+FP+FN}$$
(21)$$SE=\frac{TP}{TP+FN}$$
(22)$$SP=\frac{TN}{TN+FP}$$
(23)$$F_{1}=\frac{2TP}{2TP+FN+FP}$$

In the formula, TP and FP are true positive and false positive, respectively, which represents the correct classification and wrong classification of pixels in which the model detects blood vessels. Accordingly, TN and FN are true negative and false negative, respectively, representing being correctly and wrongly classified into the background by the model, respectively.

### 4.4. Experimental Settings

Our model is based on the Pytorch framework and trained on an RTX2060(6GB). In the training process, the Adam optimizer is used. The initial learning rates of the training are 0.0025 and 0.002 in the two datasets, respectively. The learning rates in the two datasets decay 0.8 times every 20 and 8 epochs, respectively. To avoid over-fitting of the model, we adopted L2 regularization and set the weight decay of the optimizer to 0.000007.

## 5. Results

### 5.1. Experiment of Training Strategy

In this part, we use two experiments to verify the ability of our collaborative patch training strategy in improving segmentation accuracy and saving computing resources. In terms of improving the segmentation accuracy, we compared the performance of our model with and without the collaborative patch training strategy on the two datasets to verify the effectiveness of the collaborative patch training strategy. Model MF-Net and model CPMF-Net for comparison maintain the same settings in all aspects, except in the collaborative patch training strategy. In addition, in order to improve the objectivity of the verification results, we also add the contrast experiment of the U-Net model and the U-Net model with the collaborative patch training strategy (CPU-Net). The Figure 7 shows the qualitative segmentation results of the four models. It can be found that the segmentation results of U-Net and MF-Net are inferior to others in the continuity of the vessel structure, and there are wrong segmentations of large areas. In contrast, the segmentation results of the CPU-Net model and CPMF-Net model with collaborative patch training strategy have both achieved good performances in the continuity of vessel structure and the accuracy of segmentation. The difference between the segmentation results of the same group of models reflects whether the information loss caused by patch extraction can be compensated. The segmentation results of the two baseline models after U-Net and MF-Net applying the collaborative patch training strategy show that this strategy can improve the accuracy of model segmentation by supplementing the lost information. The quantitative comparison result of the models show in Table 2. It can be found that the collaborative patch training strategy has achieved very good performance in improving the segmentation accuracy, whether in an MF-Net model or a U-Net model. Specifically, all the indicators are improved on two datasets by introducing the collaborative patch training strategy, except SP index of the STARE dataset.

As for saving computing resources, we use floating-point operations per second (Flops) and total memory occupation (Memory) indicators to make a quantitative comparison among the models based on whole-graph training, patch training and collaborative patch strategy training, respectively. Table 3 shows the quantitative comparison results among the models. It can be found that the model based on the whole-graph training (WMF-Net) has the maximum amount of floating-point calculation and the maximum computing memory requirement. Although the model based on collaborative patch strategy training has a small increase in computation and memory usage compared with the model based on single-patch training, the overall computation and memory usage still remain at a low level. Moreover, the performance of the collaborative patch training strategy improved the segmentation accuracy of the model, which means the collaborative patch training strategy can adopt smaller patch size under the same segmentation accuracy, achieving lower computational load and memory requirements, which is very helpful for the clinical application of the computer-aided diagnosis.

### 5.2. Experiment of Segmentation Model

In this sub-section, we investigate the roles of each of the two attention mechanisms, namely adaptive coordinate attention module and gated self-attention module, and their fusion in our model.

#### 5.2.1. Experiment of Adaptive Coordinate Attention Module

In this part, we first verify the effectiveness of the adaptive coordinate attention (ACA) module without using the gated self-attention (GSA) module. Then, we compare our module with other channel attention modules: (1) squeeze-and-excitation attention module (SE) and (2) classical coordinate attention module (CA). Figure 8 shows the qualitative segmentation results of the four models. It can be seen from the figure that the models with the attention mechanism such as SE, CA, or ACA have improved the segmentation results compared with Basenet, especially, our model integrated with the ACA module has achieved the best performance in the detection capability of small vessels and the continuity of vessel structure segmentation. Table 4 shows the quantitative comparison between models, where Basenet represents the model after using convolution to replace all the adaptive coordinate attention modules in the model, SE and CA represent the model after using corresponding attention modules, respectively. It can be seen from the table that compared with Basenet, the ACA module has improved in all indicators except the ACC indicator on the DRIVE dataset. Compared with the models with the SE or CA modules, the one with the ACA module has achieved the best performance in F1/ACC/AUC on two datasets, reaching 82.82%/95.67%/98.18% and 85.14%/97.07%/99.15%, respectively, which shows that our adaptive coordinate attention module is effective in improving segmentation accuracy.

#### 5.2.2. Experiment of Gated Self-Attention Module

In this part, we first verify the effectiveness of the gated self-attention module without adaptive coordinate attention. Then, in order to verify that our proposed gated self-attention module can alleviate the negative impact caused by insufficient training when there are few samples, we also compared the gated self-attention module with the Swin-Transformer (SWT) module. Figure 9 shows the comparison results of qualitative segmentation among the models. It can be seen that on the DRIVE dataset, the GSA model gives full play to the advantages of self-attention and has the best performance in the detection and segmentation of fine blood vessels. On the STARE dataset, due to the relatively small amount of data and large changes in image quality, the SW model has achieved the poorest segmentation results. In contrast, due to adaptive control, the segmentation results of the GSA model do not deteriorate significantly. The good segmentation results on the two datasets show the effectiveness of the self-attention module of our gated self-attention module. Table 5 shows the quantitative comparison results among three variants of our model, where Basenet and SW are the models where a convolutional block and Swin self-attention block are used in place of the gated self-attention module, respectively. The three models have the same settings as that of the Basenet, except the adding module to be substituted in different models. It can be seen from the Table 5 that the indicators of SW on both datasets have decreased compared with that of Basenet, which is caused by inadequate training of SWT on smaller datasets. In contrast, the GSA we designed has the ability to mitigate the negative impact of inadequate training, so it has achieved the best performance on both datasets.

#### 5.2.3. Ablation Experiment

In order to verify the futher effectiveness of the combination of the two attention modules, we conducted ablation experiments. The quantitative results are shown in Table 6. It can be found that the integration of the two attention modules improves the ability of the model to obtain specific features, and when the two attention modules are integrated with each other at the same time, the model can combine the advantages of the two modules to implement better segmentation.

### 5.3. Comparison with the State-of-the-Art Methods

In this part, we compare our method with the other nine advanced methods. Figure 10 intuitively shows the segmentation results of our model and some advanced models. It can be seen from the figure that the segmentation results of our model are the closest to the ground truth marked by experts, and our model CPMF-Net has the best performance in terms of segmentation integrity and continuity. The results of quantitative comparison are shown in Table 7. It can be found that our method achieves the best performance in three of all five indicators, including F1, ACC and AUC on DRIVE dataset, reaching 82.94%, 95.78% and 98.19%, respectively. Meanwhile, our method also achieves the best performance in F1, SE, ACC and AUC indicators on the STARE dataset, and the SE indicator among them is 2.49% higher than that of CSU-Net, which ranks second. Figure 11 shows the ROC curves of our model and some advanced models, from which we can find that our model outperforms the other models on the ROC curve, reflecting the overall performance of segmentation.

In addition, considering the importance of generalization ability to the practical application of the method [22,23], we also evaluated the generalization ability of our method through cross-training. Table 8 shows the quantitative comparison of the generalization performance between our method and the other three existing methods. It can be found that our method has better generalization capability than other methods on both datasets. It has achieved the optimal performance of all indicators on the DRIVE dataset and the optimal performance on the STARE dataset, except for the SP indicator.

**Table 7 sensors-22-09210-t007:** Comparison with the most advanced methods on DRIVE and STARE.

Model	Year	DRIVE					STARE				
		F1 (%)	SE (%)	SP (%)	ACC (%)	AUC (%)	F1 (%)	SE (%)	SP (%)	ACC (%)	AUC (%)
U-Net [8]	2015	81.36	77.92	98.12	95.61	97.66	83.27	82.95	98.15	96.60	98.76
R2U-Net [24]	2018	78.07	83.05	95.86	94.27	95.95	77.58	79.62	97.08	95.30	97.17
CE-Net [25]	2019	78.64	77.78	97.21	94.80	97.11	82.74	84.04	97.83	96.42	98.68
Xu et al. [26]	2020	82.52	79.53	98.07	95.57	98.04	83.08	83.78	97.41	95.90	98.17
Zhou et al. [27]	2020	80.35	74.73	**98.35**	95.35	97.13	81.32	77.76	98.32	96.05	97.40
Li et al. [28]	2021	-	79.21	98.10	95.68	98.06	-	83.52	98.23	96.78	98.75
CSU-Net [29]	2021	82.51	80.71	98.01	95.65	98.01	85.16	84.32	98.45	97.02	98.25
Bridge-Net [30]	2022	82.03	78.53	98.18	95.65	98.34	82.89	80.02	**98.64**	96.68	99.01
Li et al. [31]	2022	82.88	**83.59**	97.31	95.71	98.10	83.63	83.52	98.23	96.71	98.75
CPMF-Net(ours)	2022	**82.94**	83.54	97.53	**95.78**	**98.19**	**85.66**	**86.81**	98.20	**97.03**	**99.16**

The excellent performance in SE and AUC indicators means that our proposed method has stronger vessel perception ability while maintaining segmentation accuracy. This ability enables the model to better detect and segment fine vessel structures, which is very helpful for the computer-aided diagnosis in the clinical diagnosing application of early ocular diseases.

## 6. Conclusions

In this paper, according to the characteristics of the retinal blood vessel segmentation task, we propose a new retinal blood vessel segmentation method, namely the multi-feature segmentation method based on collaborative patches. By combining a powerful multi-feature network with an effective collaborative patch training strategy, high-precision segmentation without the extremely rigid hardware condition can be achieved. The experimental results on the DRIVE and STARE datasets show the effectiveness and great application potential of our approach. For the future work, we plan to optimize the feature extraction ability of the segmentation model and the associated information compensation ability of the collaborative patch training strategy.

## Figures and Tables

**Figure 1 sensors-22-09210-f001:**
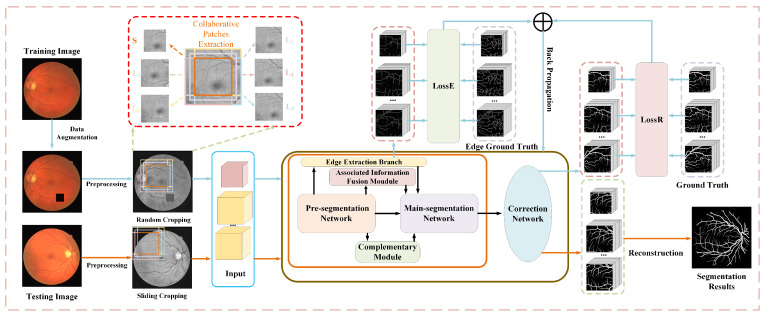
Flow chart of the proposed method. The gray dotted box shows the patch extraction process. The extracted small patch corresponds to the target area, and each of the large patches corresponds to the combination of the target area and some global information.

**Figure 2 sensors-22-09210-f002:**
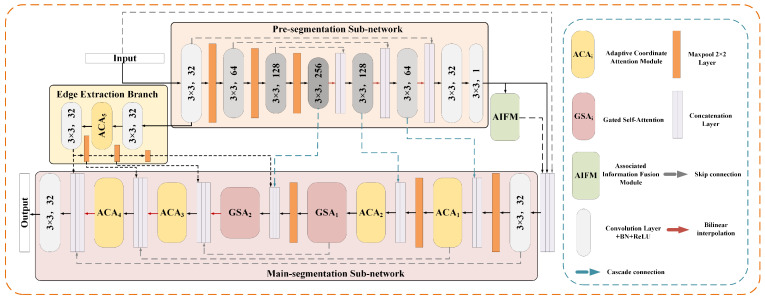
Our proposed multi-feature segmentation network (MF-Net). The 3 × 3 and numbers (1, 32, 64, 128 and 256) in the gray rounded rectangle correspond to the size of the convolution kernel and the number of output channels, respectively. The network consists of the pre-segmentation network (orange box), the main-segmentation network (pink box) and the edge extraction branch (yellow box).

**Figure 3 sensors-22-09210-f003:**
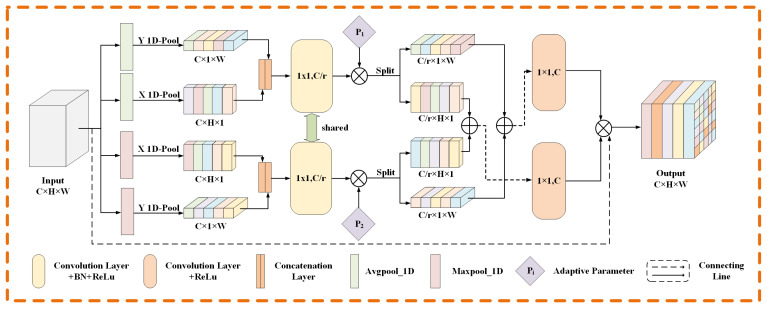
Details of the designed adaptive coordinate attention module. The 1 × 1 and numbers (C, C/r) in the Convolution block correspond to the size of the convolution kernel and the number of output channels, respectively. Additional maxpooling coding is used to elevate coding differences between capillaries and background. Two learnable parameter weights are used to adaptively combine the two coding methods.

**Figure 4 sensors-22-09210-f004:**
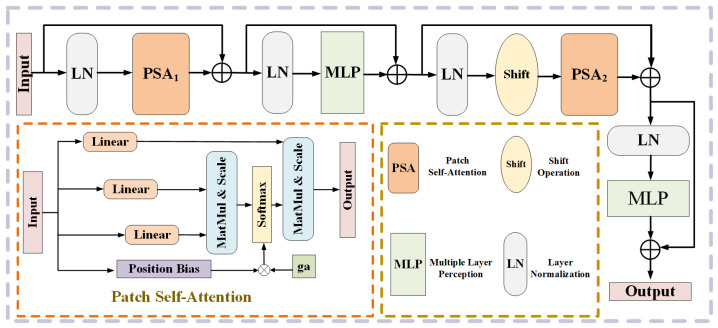
Details of the designed gated self-attention module. The calculation details of patch self-attention are shown in the orange dotted line box.

**Figure 5 sensors-22-09210-f005:**
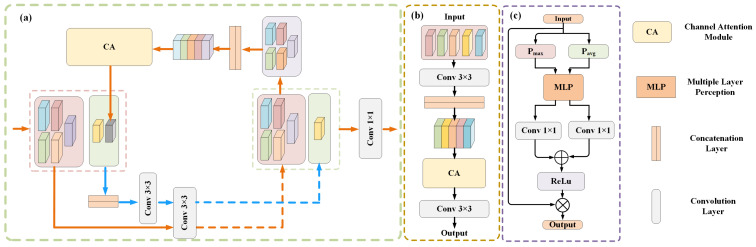
Details of the associated information fusion module and the associated information correction network. (**a**) An associated information correction sub-network for correcting the global information following the semantic segmentation network. (**b**) The channel attention module used in the associated correction network and associated fusion module. (**c**) An associated information fusion module used to supplement global associated information between two-level networks.

**Figure 6 sensors-22-09210-f006:**
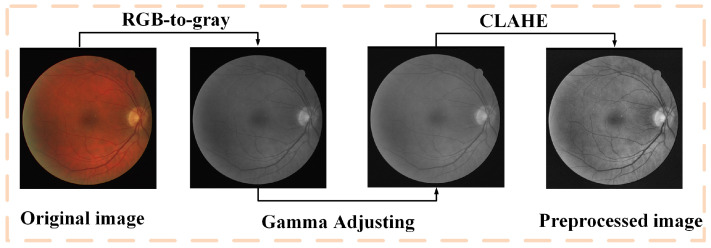
Details of the pre-processing process.

**Figure 7 sensors-22-09210-f007:**
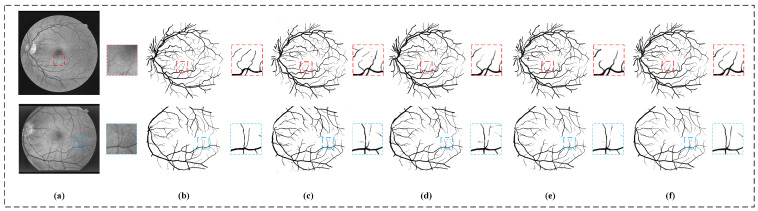
Model segmentation results using different training strategies. The top row is based on DRIVE datasets, and the bottom row is STARE datasets. (**a**) The original image, (**b**) the ground truth, (**c**) the segmentation result of U-Net, (**d**) the segmentation result of CPU-Net, (**e**) the segmentation result of MF-Net and (**f**) the segmentation result of our CPMF-Net.

**Figure 8 sensors-22-09210-f008:**
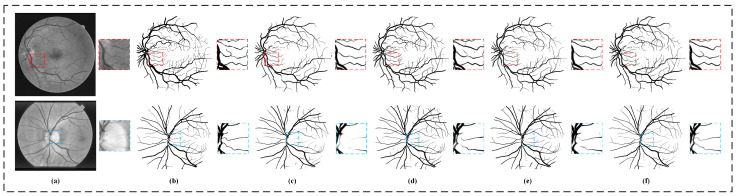
The model segmentation results with different channel attention modules integrated. The top row is the result on the DRIVE dataset, and the bottom row is the one on the STARE dataset. (**a**) The original image, (**b**) the ground truth, (**c**) the segmentation result of Basenet, (**d**) the segmentation result integrated with SE module, (**e**) the segmentation result integrated with CA module and (**f**) the segmentation result integrated with ACA.

**Figure 9 sensors-22-09210-f009:**
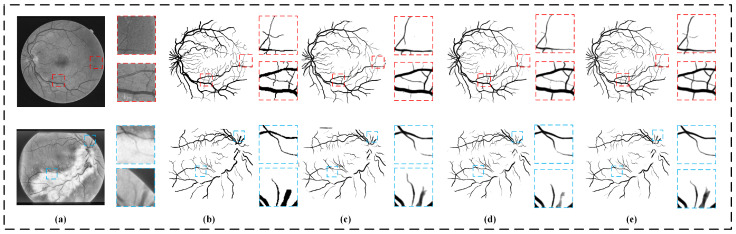
The model segmentation results of different self-attention modules integrated. The top row is the based on DRIVE dataset, and the bottom row is the STARE dataset. (**a**) The original image, (**b**) the ground truth, (**c**) the segmentation result of Basenet, (**d**) the segmentation result of the integrated SW module and (**e**) the segmentation result of the our model integrated with the GSA module.

**Figure 10 sensors-22-09210-f010:**
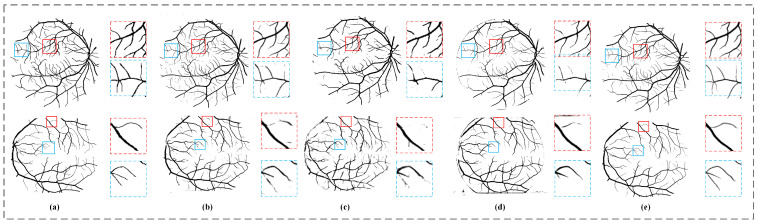
Comparison of segmentation results between our approach and some other advanced approaches. (**a**) The ground truth, (**b**) the segmentation result of UNet model, (**c**) the segmentation result of R2Unet model, (**d**) the segmentation result of CE-Net model and (**e**) the segmentation result of our model.

**Figure 11 sensors-22-09210-f011:**
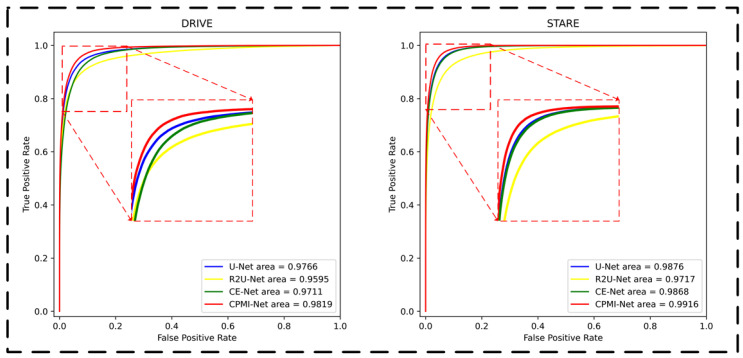
ROC curves of our model and some advanced models.

**Table 1 sensors-22-09210-t001:** Dataset information.

Dataset	DRIVE	STARE
Number of Images	40	20
Original Size	584 × 565	700 × 605
Patch Size	72 × 72	72 × 72
Tran/Test Split	20/20	16/4

**Table 2 sensors-22-09210-t002:** Quantitative comparison of the performance of different training methods in segmentation accuracy.

Model	DRIVE					STARE				
	F1 (%)	SE (%)	SP (%)	ACC (%)	AUC (%)	F1 (%)	SE (%)	SP (%)	ACC (%)	AUC (%)
U-Net	81.36	77.92	**98.12**	95.61	97.66	83.27	82.95	98.15	96.60	98.76
CPU-Net	82.55	82.77	97.54	95.70	98.07	84.67	86.21	98.02	96.81	98.94
MF-Net	82.10	82.19	97.50	95.59	97.84	83.64	80.77	**98.60**	96.78	98.84
CPMF-Net	**82.94**	**83.45**	97.53	**95.78**	**98.19**	**85.66**	**86.81**	98.20	**97.03**	**99.16**

**Table 3 sensors-22-09210-t003:** Quantitative comparison of the computational resource occupation among different training methods.

Model	Flops	Memory
MF-Net	2.04 G	38.43 M
CPMF-Net	2.05 G	41.25 M
WMF-Net	141.51 G	2651 M

**Table 4 sensors-22-09210-t004:** Quantitative performance of different channel attention modules in retinal segmentation task.

Model	DRIVE			STARE		
	F1 (%)	ACC (%)	AUC (%)	F1 (%)	ACC (%)	AUC (%)
Basenet	82.03	95.74	98.06	84.84	97.01	99.13
SE	82.49	95.60	98.11	84.95	96.72	99.12
CA	82.46	95.62	98.14	84.66	96.59	**99.15**
ACA	**82.82**	**95.67**	**98.18**	**85.14**	**97.07**	**99.15**

**Table 5 sensors-22-09210-t005:** Quantitative performances of the models based on different self-attention modules in retinal segmentation task.

Model	DRIVE			STARE		
	F1 (%)	ACC (%)	AUC (%)	F1 (%)	ACC (%)	AUC (%)
Basenet	82.03	95.74	98.06	84.84	97.01	99.13
SW	81.54	95.73	98.11	84.86	96.97	99.07
GSA	**82.04**	**95.75**	**98.15**	**85.27**	**97.15**	**99.14**

**Table 6 sensors-22-09210-t006:** Ablation experiments on DRIVE and STARE. Basenet represents the basic model using convolutional blocks and without the two additional attention modules.

Model	DRIVE			STARE		
	F1 (%)	ACC (%)	AUC (%)	F1 (%)	ACC (%)	AUC (%)
Basenet	82.03	95.74	98.06	84.84	97.01	99.13
Basenet + GSA	82.04	95.75	98.15	85.27	**97.15**	99.14
Basenet + ACA	82.82	95.67	98.18	85.14	97.07	99.15
Basenet + GSA + ACA	**82.94**	**95.78**	**98.19**	**85.66**	97.03	**99.16**

**Table 8 sensors-22-09210-t008:** Cross-training results on DRIVE and STARE datasets.

Test Set	Training Set	Model	SE	SP	ACC	AUC
STARE	DRIVE	Fraz [22]	72.42	97.92	94.56	96.97
		Li [23]	72.73	98.10	94.86	96.77
		Yan [32]	72.92	**98.15**	94.94	95.99
		CPMF-Net(ours)	**75.93**	**98.15**	**95.39**	**97.53**
DRIVE	STARE	Fraz [22]	70.10	97.70	94.95	96.71
		Li [23]	70.27	98.28	95.45	96.71
		Yan [32]	72.11	**98.40**	95.69	97.08
		CPMF-Net(ours)	**80.24**	98.12	**96.04**	**98.51**

## Data Availability

The datasets used in this work are publicly available online.

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
