# Peer review of "CPMF-Net: Multi-Feature Network Based on Collaborative Patches for Retinal Vessel Segmentation"

_sensors, 2022, doi:10.3390/s22239210_

Round 1
Reviewer 1 Report
This article proposed a new multi-information network for retinal vessel segmentation using collaborative patches structure, which shows better performance than other existing methods. In my opinion, their results are very interesting and important for researchers, and the proposed network promises a future application in the segmentation of vessel imaging. Therefore, I suggest this article to be published in Sensors after minor revision and answering some small questions.
1. About the training of the network, how long it will take to train the whole model? The data augmentation was used during training, and how much images are used for the training after data augmentation?
2. What about overfitting? Was the testing image from the same kind of dataset or not? Considering that the training data is kind of small, it would be better to add some discussion about the generalization of this trained network for other type of datasets.
3. There are some typos in the paper. For example, ‘The module is composed of pacth self-attention …’ in page 6; ‘3.4.2. associated Information Transmission’ in page 7 should be capital ‘Associated…’
Reviewer 2 Report
In this work, the authors proposed a new image segmentation method, combining two neural networks at different length scales to improve segmentation accuracy. For the network design, the authors proposed an attention module “adaptive coordinate attention module” to extract directional information. Another module “gated self-attention module” was used to increase the efficiency of model. To prevent missing information in the patched dataset, the authors proposed a “collaborative patch training strategy”, including small patch with interested region and five large patches with its surrounding information. The proposed network design was validated in DRIVE and STARE dataset. Overall, this is an interesting paper, and the authors proposed a few new ideas for improving model performance and efficiency. There are some points are not clear and need further clarification. I recommend a minor revision on this manuscript.
(1). How did the authors optimize two segmentation networks together with edge extraction branch?
(2). The one-dimensional pooling is a smart way of identifying the direction information. However, does it work when handling rotated or tilted images in real applications?
(3). It is not clear what regularization parameters did the authors setup?
Reviewer 3 Report
The paper proposes a new method for retinal vessel segmentation. The paper is well written but there are many typos that need to be corrected. The authors figures help a lot to understand the complex architecture of the method, some minor topics need further extension.
The authors offer metrics for individual modules and variations od the proposed method. Furthermore, in table 7, they present chronological comparison with other methods. The comparison seems fair, and the detail of the paper is enough to ensure other practitioners to repeat the experiments.
Examples in figure 9, 10 shown the capabilities of the new method.
I recommend the total review of the paper to find the typos in the text, but the writing style is ok. Some acronyms are not defined.
Specific suggestions:
Line 19 – “…, more than 70% information to the brain is provided via eyes …”, please cite.
Line 25 - “According to medical research, the appearance of retinal diseases …”, please cite.
Line 73 – You argue that CNN can automatically learn to recognize features, compared to SVM or KNN. I agree that CNN are suited for image processing, by avoiding the dimensionality curse, but theoretically SVM and KNN can learn the same problems. On the other hand, the word “automacally” can be missunderstood like a self-learning algorithm. Please rewrite this sentence.
Line 129 – The name of the network could be misleading, for example, one can argue that a multi-information network is a neural net that take exogenous information. However, the name is well presented in the paper, so changing the name is a minor suggestion.
The figures are well made, and they really improve the paper reading. I suggest including what dashed arrow mean. Thank you for including Figure 3, it is better than the figure of the original ACA paper.
After equation (6) you defined the concatenation notation but there is “[.]” instead “[,]”
Figure 3, the parameter r in the 1x1 convolutional its not defined, Is C/r always an integer? if it is not, I suggest using brackets for floor
Line 149 – You explain that the B matrix value comes from the deviation matrix B hat, but it was not defined, I suggest adding a cite, referencing this process.
Line 158 – I suggest to extent this subsection, it is hard to understand how the collaborative patch works and it is one of the main contributions of the paper.
Line 171 – capitalize “associated”
Figure 6 – CLAHE is not defined
Figure 7 – I suggest expanding the figure size to the left (in the same way of table 2) to get a better detail.
Table 2 – I suggest adding a vertical line to separate DRIVE and STARE results, please do this for all tables.
Figure 10 – Same suggestion of figure 7
Line 205 – Add AUC definition.
